# Recovery of Coherent Data via Low-Rank Dictionary Pursuit

**Guangcan Liu**
Department of Statistics and Biostatistics
Department of Computer Science
Rutgers University
Piscataway, NJ 08854, USA
gcliu@rutgers.edu

**Ping Li**
Department of Statistics and Biostatistics
Department of Computer Science
Rutgers University
Piscataway, NJ 08854, USA
pingli@rutgers.edu

## Abstract

The recently established RPCA [4] method provides a convenient way to restore low-rank matrices from grossly corrupted observations. While elegant in theory and powerful in reality, RPCA is not an ultimate solution to the low-rank matrix recovery problem. Indeed, its performance may not be perfect even when data are strictly low-rank. This is because RPCA ignores clustering structures of the data which are ubiquitous in applications. As the number of cluster grows, the coherence of data keeps increasing, and accordingly, the recovery performance of RPCA degrades. We show that the challenges raised by coherent data (i.e., data with high coherence) could be alleviated by Low-Rank Representation (LRR) [13], provided that the dictionary in LRR is configured appropriately. More precisely, we mathematically prove that if the dictionary itself is low-rank then LRR is immune to the coherence parameter which increases with the underlying cluster number. This provides an elementary principle for dealing with coherent data and naturally leads to a practical algorithm for obtaining proper dictionaries in unsupervised environments. Experiments on randomly generated matrices and real motion sequences verify our claims. *See the full paper at arXiv:1404.4032.*

## 1 Introduction

Nowadays our data are often high-dimensional, massive and full of gross errors (e.g., corruptions, outliers and missing measurements). In the presence of gross errors, the classical Principal Component Analysis (PCA) method, which is probably the most widely used tool for data analysis and dimensionality reduction, becomes brittle — A single gross error could render the estimate produced by PCA arbitrarily far from the desired estimate. As a consequence, it is crucial to develop new statistical tools for robustifying PCA. A variety of methods have been proposed and explored in the literature over several decades, e.g., [2, 3, 4, 8, 9, 10, 11, 12, 24, 13, 16, 19, 25]. One of the most exciting methods is probably the so-called RPCA (Robust Principal Component Analysis) method [4], which was built upon the exploration of the following low-rank matrix recovery problem:

**Problem 1 (Low-Rank Matrix Recovery)** *Suppose we have a data matrix $X \in \mathbb{R}^{m \times n}$ and we know it can be decomposed as*

$$X = L_0 + S_0, \tag{1.1}$$

*where $L_0 \in \mathbb{R}^{m \times n}$ is a low-rank matrix each column of which is a data point drawn from some low-dimensional subspace, and $S_0 \in \mathbb{R}^{m \times n}$ is a sparse matrix supported on $\Omega \subseteq \{1, \cdots, m\} \times \{1, \cdots, n\}$. Except these mild restrictions, both components are arbitrary. The rank of $L_0$ is unknown, the support set $\Omega$ (i.e., the locations of the nonzero entries of $S_0$) and its cardinality (i.e., the amount of the nonzero entries of $S_0$) are unknown either. In particular, the magnitudes of the nonzero entries in $S_0$ may be arbitrarily large. Given $X$, can we recover both $L_0$ and $S_0$, in a scalable and exact fashion?*

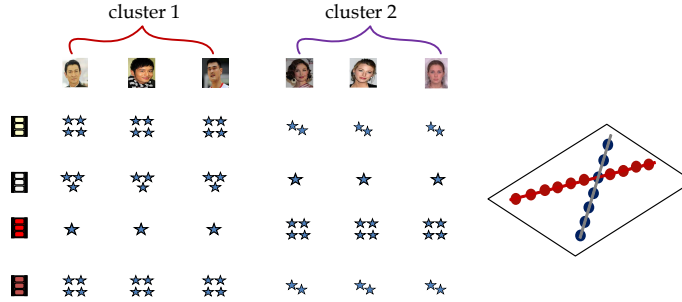

Figure 1: Exemplifying the extra structures of low-rank data. Each entry of the data matrix is a grade that a user assigns to a movie. It is often the case that such data are low-rank, as there exist wide correlations among the grades that different users assign to the same movie. Also, such data could own some clustering structure, since the preferences of the same type of users are more similar to each other than to those with different gender, personality, culture and education background. In summary, such data (1) are often low-rank and (2) exhibit some clustering structure.

The theory of RPCA tells us that, very generally, when the low-rank matrix $L_0$ is meanwhile *incoherent* (i.e., with low coherence), both the low-rank and the sparse matrices can be *exactly* recovered by using the following convex, potentially scalable program:

$$\min_{L,S} \|L\|_* + \lambda \|S\|_1, \quad \text{s.t.} \quad X = L + S, \tag{1.2}$$

where $\| \cdot \|_*$ is the nuclear norm [7] of a matrix, $\| \cdot \|_1$ denotes the $\ell_1$ norm of a matrix seen as a long vector, and $\lambda > 0$ is a parameter. Besides of its elegance in theory, RPCA also has good empirical performance in many practical areas, e.g., image processing [26], computer vision [18], radar imaging [1], magnetic resonance imaging [17], etc.

While complete in theory and powerful in reality, RPCA cannot be an ultimate solution to the low-rank matrix recovery Problem 1. Indeed, the method might not produce perfect recovery even when $L_0$ is strictly low-rank. This is because RPCA captures only the low-rankness property, which however is not the only property of our data, but essentially ignores the *extra structures* (beyond low-rankness) widely existing in data: Given the low-rankness constraint that the data points (i.e., columns vectors of $L_0$) locate on a low-dimensional subspace, it is unnecessary for the data points to locate on the subspace *uniformly at random* and it is quite normal that the data may have some extra structures, which specify in more detail *how* the data points locate on the subspace. Figure 1 demonstrates a typical example of extra structures; that is, the clustering structures which are ubiquitous in modern applications. Whenever the data are exhibiting some clustering structures, RPCA is no longer a method of perfection. Because, as will be shown in this paper, while the rank of $L_0$ is fixed and the underlying cluster number goes large, the coherence of $L_0$ keeps heightening and thus, arguably, the performance of RPCA drops.

To better handle coherent data (i.e., the cases where $L_0$ has large coherence parameters), a seemingly straightforward idea is to *avoid* the coherence parameters of $L_0$. However, as explained in [4], the coherence parameters are indeed necessary (if there is no additional condition assumed on the data). This paper shall further indicate that the coherence parameters are related in nature to some extra structures intrinsically existing in $L_0$ and therefore cannot be discarded simply. Interestingly, we show that it is possible to *avoid* the coherence parameters by using some additional conditions, which are easy to obey in supervised environment and can also be approximately achieved in unsupervised environment. Our study is based on the following convex program termed Low-Rank Representation (LRR) [13]:

$$\min_{Z,S} \|Z\|_* + \lambda \|S\|_1, \quad \text{s.t.} \quad X = AZ + S, \tag{1.3}$$

where $A \in \mathbb{R}^{m \times d}$ is a size-$d$ dictionary matrix constructed in advance[1], and $\lambda > 0$ is a parameter. In order for LRR to avoid the coherence parameters which increase with the cluster number underlying

$L_0$, we prove that it is sufficient to construct in advance a dictionary $A$ which is low-rank by itself. This gives a generic prescription to defend the possible infections raised by coherent data, providing an elementary criteria for learning the dictionary matrix $A$. Subsequently, we propose a simple and effective algorithm that utilizes the output of RPCA to construct the dictionary in LRR. Our extensive experiments demonstrated on randomly generated matrices and motion data show promising results. In summary, the contributions of this paper include the following:

⋄ For the first time, this paper studies the problem of recovering low-rank, and coherent (or less incoherent as equal) matrices from their corrupted versions. We investigate the physical regime where coherent data arise. For example, the widely existing clustering structures may lead to coherent data. We prove some basic theories for resolving the problem, and also establish a practical algorithm that outperforms RPCA in our experimental study.

⋄ Our studies help reveal the *physical* meaning of coherence, which is now standard and widely used in various literatures, e.g., [2, 3, 4, 25, 15]. We show that the coherence parameters are not "assumptions" for a proof, but rather some excellent quantities that relate in nature to the *extra structures* (beyond low-rankness) intrinsically existing in $L_0$.

⋄ This paper provides insights regarding the LRR model proposed by [13]. While the special case of $A = X$ has been extensively studied, the LRR model (1.3) with general dictionaries is not fully understood yet. We show that LRR (1.3) equipped with proper dictionaries could well handle coherent data.

⋄ The idea of replacing $L$ with $AZ$ is essentially related to the spirit of matrix factorization which has been explored for long, e.g., [20, 23]. In that sense, the explorations of this paper help to understand why factorization techniques are useful.

## 2   Summary of Main Notations

Capital letters such as $M$ are used to represent matrices, and accordingly, $[M]_{ij}$ denotes its $(i,j)$th entry. Letters $U$, $V$, $\Omega$ and their variants (complements, subscripts, etc.) are reserved for left singular vectors, right singular vectors and support set, respectively. We shall abuse the notation $U$ (resp. $V$) to denote the linear space spanned by the columns of $U$ (resp. $V$), i.e., the column space (resp. row space). The projection onto the column space $U$, is denoted by $\mathcal{P}_U$ and given by $\mathcal{P}_U(M) = UU^T M$, and similarly for the row space $\mathcal{P}_V(M) = MVV^T$. We shall also abuse the notation $\Omega$ to denote the linear space of matrices supported on $\Omega$. Then $\mathcal{P}_\Omega$ and $\mathcal{P}_{\Omega^\perp}$ respectively denote the projections onto $\Omega$ and $\Omega^c$ such that $\mathcal{P}_\Omega + \mathcal{P}_{\Omega^\perp} = \mathcal{I}$, where $\mathcal{I}$ is the identity operator. The symbol $(\cdot)^+$ denotes the Moore-Penrose pseudoinverse of a matrix: $M^+ = V_M \Sigma_M^{-1} U_M^T$ for a matrix $M$ with Singular Value Decomposition (SVD)[2] $U_M \Sigma_M V_M^T$.

Six different matrix norms are used in this paper. The first three norms are functions of the singular values: 1) The operator norm (i.e., the largest singular value) denoted by $\|M\|$, 2) the Frobenius norm (i.e., square root of the sum of squared singular values) denoted by $\|M\|_F$, and 3) the nuclear norm (i.e., the sum of singular values) denoted by $\|M\|_*$. The other three are the $\ell_1$, $\ell_\infty$ (i.e., sup-norm) and $\ell_{2,\infty}$ norms of a matrix: $\|M\|_1 = \sum_{i,j} |[M]_{ij}|$, $\|M\|_\infty = \max_{i,j}\{|[M]_{ij}|\}$ and $\|M\|_{2,\infty} = \max_j\{\sqrt{\sum_i [M]_{ij}^2}\}$, respectively.

The Greek letter $\mu$ and its variants (e.g., subscripts and superscripts) are reserved for the coherence parameters of a matrix. We shall also reserve two lower case letters, $m$ and $n$, to respectively denote the data dimension and the number of data points, and we use the following two symbols throughout this paper:

$$n_1 = \max(m,n) \quad \text{and} \quad n_2 = \min(m,n).$$

## 3   On the Recovery of Coherent Data

In this section, we shall firstly investigate the physical regime that raises coherent (or less incoherent) data, and then discuss the problem of recovering coherent data from corrupted observations, providing some basic principles and an algorithm for resolving the problem.

### 3.1 Coherence Parameters and Their Properties

As the rank function cannot fully capture all characteristics of $L_0$, it is necessary to define some quantities to measure the effects of various extra structures (beyond low-rankness) such as the clustering structure as demonstrated in Figure 1. The *coherence* parameters defined in [3, 4] are excellent exemplars of such quantities.

#### 3.1.1 Coherence Parameters: $\mu_1, \mu_2, \mu_3$

For an $m \times n$ matrix $L_0$ with rank $r_0$ and SVD $L_0 = U_0 \Sigma_0 V_0^T$, some important properties can be characterized by three coherence parameters, denoted as $\mu_1, \mu_2$ and $\mu_3$, respectively. The first coherence parameter, $1 \leq \mu_1(L_0) \leq m$, which characterizes the column space identified by $U_0$, is defined as

$$\mu_1(L_0) = \frac{m}{r_0} \max_{1 \leq i \leq m} \|U_0^T e_i\|_2^2, \tag{3.4}$$

where $e_i$ denotes the $i$th standard basis. The second coherence parameter, $1 \leq \mu_2(L_0) \leq n$, which characterizes the row space identified by $V_0$, is defined as

$$\mu_2(L_0) = \frac{n}{r_0} \max_{1 \leq j \leq n} \|V_0^T e_j\|_2^2. \tag{3.5}$$

The third coherence parameter, $1 \leq \mu_3(L_0) \leq mn$, which characterizes the joint space identified by $U_0 V_0^T$, is defined as

$$\mu_3(L_0) = \frac{mn}{r_0} (\|U_0 V_0^T\|_\infty)^2 = \frac{mn}{r_0} \max_{i,j} (|\langle U_0^T e_i, V_0^T e_j \rangle|)^2. \tag{3.6}$$

The analysis in RPCA [4] merges the above three parameters into a single one: $\mu(L_0) = \max\{\mu_1(L_0), \mu_2(L_0), \mu_3(L_0)\}$. As will be seen later, the behaviors of those three coherence parameters are different from each other, and hence it is more adequate to consider them individually.

#### 3.1.2 $\mu_2$-phenomenon

According to the analysis in [4], the success condition (regarding $L_0$) of RPCA is

$$\text{rank}(L_0) \leq \frac{c_r n_2}{\mu(L_0)(\log n_1)^2}, \tag{3.7}$$

where $\mu(L_0) = \max\{\mu_1(L_0), \mu_2(L_0), \mu_3(L_0)\}$ and $c_r > 0$ is some numerical constant. Thus, RPCA will be less successful when the coherence parameters are considerably larger. In this subsection, we shall show that the widely existing clustering structure can enlarge the coherence parameters and, accordingly, downgrades the performance of RPCA.

Given the restriction that $\text{rank}(L_0) = r_0$, the data points (i.e., column vectors of $L_0$) are unnecessarily sampled from a $r_0$-dimensional subspace *uniformly at random*. A more realistic interpretation is to consider the data points as samples from the union of $k$ number of subspaces (i.e., clusters), and the sum of those multiple subspaces together has a dimension $r_0$. That is to say, there are multiple "small" subspaces inside one $r_0$-dimensional "large" subspace, as exemplified in Figure 1. Whenever the low-rank matrix $L_0$ is meanwhile exhibiting such clustering behaviors, the second coherence parameter $\mu_2(L_0)$ (and so $\mu_3(L_0)$) will increase with the number of clusters underlying $L_0$, as shown in Figure 2. When the coherence is heightening, (3.7) suggests that the performance of RPCA will drop, as verified in Figure 2(d). Note here that the variation of $\mu_3$ is mainly due to the variation of the row space, which is characterized by $\mu_2$. We call the phenomena shown in Figure 2(b)∼(d) as the "$\mu_2$-phenomenon". Readers can also refer to the full paper to see why the second coherence parameter increases with the cluster number underlying $L_0$.

Interestingly, one may have noticed that $\mu_1$ is invariant to the variation of the clustering number, as can be seen from Figure 2(a). This is because the clustering behavior of the data points can only affect the row space, while $\mu_1$ is defined on the column space. Yet, if the row vectors of $L_0$ also own some clustering structure, $\mu_1$ could be large as well. Such kind of data can exist widely in text documents and we leave this as future work.

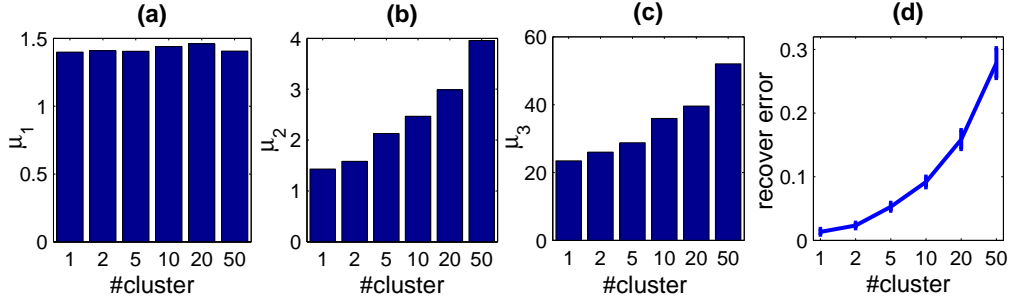

Figure 2: Exploring the influence of the cluster number, using randomly generated matrices. The size and rank of $L_0$ are fixed to be $500 \times 500$ and 100, respectively. The underlying cluster number varies from 1 to 50. For the recovery experiments, $S_0$ is fixed as a sparse matrix with 13% nonzero entries. (a) The first coherence parameter $\mu_1(L_0)$ vs cluster number. (b) $\mu_2(L_0)$ vs cluster number. (c) $\mu_3(L_0)$ vs cluster number. (d) Recover error (produced by RPCA) vs cluster number. The numbers shown in these figure are averaged from 100 random trials. The recover error is computed as $\|\hat{L}_0 - L_0\|_F / \|L_0\|_F$, where $\hat{L}_0$ denotes an estimate of $L_0$.

### 3.2 Avoiding $\mu_2$ by LRR

The $\mu_2$-phenomenon implies that the second coherence parameter $\mu_2$ is related in nature to some intrinsic structures of $L_0$ and thus cannot be eschewed without using additional conditions. In the following, we shall figure out under what conditions the second coherence parameter $\mu_2$ (and $\mu_3$) can be avoided such that LRR could well handle coherent data.

**Main Result:** We show that, when the dictionary $A$ itself is low-rank, LRR is able to avoid $\mu_2$. Namely, the following theorem is proved without using $\mu_2$. See the full paper for a detailed proof.

**Theorem 1 (Noiseless)** *Let $A \in \mathbb{R}^{m \times d}$ with SVD $A = U_A \Sigma_A V_A^T$ be a column-wisely unit-normed (i.e., $\|Ae_i\|_2 = 1, \forall i$) dictionary matrix which satisfies $\mathcal{P}_{U_A}(U_0) = U_0$ (i.e., $U_0$ is a subspace of $U_A$). For any $0 < \epsilon < 0.5$ and some numerical constant $c_a > 1$, if*

$$\text{rank}(L_0) \le \text{rank}(A) \le \frac{\epsilon^2 n_2}{c_a \mu_1(A) \log n_1} \quad and \quad |\Omega| \le (0.5 - \epsilon)mn, \tag{3.8}$$

*then with probability at least $1 - n_1^{-10}$, the optimal solution to the LRR problem* (1.3) *with $\lambda = 1/\sqrt{n_1}$ is unique and exact, in a sense that*

$$Z^* = A^+ L_0 \quad and \quad S^* = S_0,$$

*where $(Z^*, S^*)$ is the optimal solution to* (1.3).

It is worth noting that the restriction $\text{rank}(L_0) \le O(n_2 / \log n_1)$ is looser than that of PRCA[3], which requires $\text{rank}(L_0) \le O(n_2/(\log n_1)^2)$. The requirement of column-wisely unit-normed dictionary (i.e., $\|Ae_i\|_2 = 1, \forall i$) is purely for complying the parameter estimate of $\lambda = 1/\sqrt{n_1}$, which is consistent with RPCA. The condition $\mathcal{P}_{U_A}(U_0) = U_0$, i.e., $U_0$ is a subspace of $U_A$, is indispensable if we ask for exact recovery, because $\mathcal{P}_{U_A}(U_0) = U_0$ is implied by the equality $AZ^* = L_0$. This necessary condition, together with the low-rankness condition, provides an elementary criterion for learning the dictionary matrix $A$ in LRR. Figure 3 presents an example, which further confirms our main result; that is, LRR is able to avoid $\mu_2$ as long as $U_0 \subset U_A$ and $A$ is low-rank. It is also worth noting that it is unnecessary for $A$ to satisfy $U_A = U_0$, and that LRR is actually tolerant to the "errors" possibly existing in the dictionary.

The program (1.3) is designed for the case where the uncorrupted observations are noiseless. In reality this assumption is often not true and all entries of $X$ can be contaminated by a small amount of noises, i.e., $X = L_0 + S_0 + N$, where $N$ is a matrix of dense Gaussian noises. In this case, the formula of LRR (1.3) need be modified to

$$\min_{Z,S} \|Z\|_* + \lambda \|S\|_1, \quad \text{s.t.} \quad \|X - AZ - S\|_F \le \varepsilon, \tag{3.9}$$

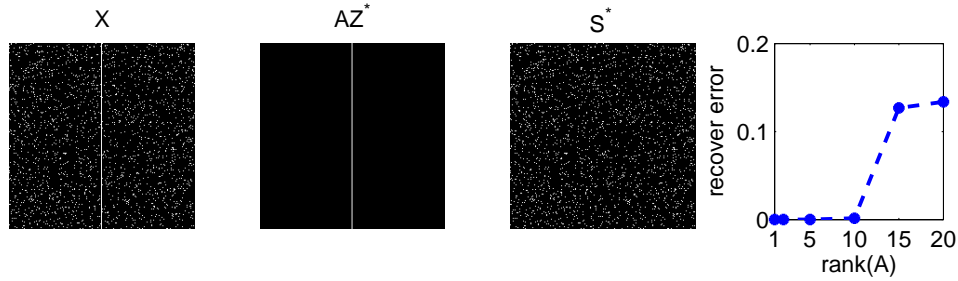

Figure 3: Exemplifying that LRR can void $\mu_2$. In this experiment, $L_0$ is a $200 \times 200$ rank-1 matrix with one column being $\mathbf{1}$ (i.e., a vector of all ones) and everything else being zero. Thus, $\mu_1(L_0) = 1$ and $\mu_2(L_0) = 200$. The dictionary is set as $A = [\mathbf{1}, W]$, where $W$ is a $200 \times p$ random Gaussian matrix (with varying $p$). As long as $\mathrm{rank}\,(A) = p+1 \le 10$, LRR with $\lambda = 0.08$ can exactly recover $L_0$ from a grossly corrupted observation matrix $X$.

where $\varepsilon$ is a parameter that measures the noise level of data. In the experiments of this paper, we consistently set $\varepsilon = 10^{-6}\|X\|_F$. In the presence of dense noises, the latent matrices, $L_0$ and $S_0$, cannot be exactly restored. Yet we have the following theorem to guarantee the near recovery property of the solution produced by the program (3.9):

**Theorem 2 (Noisy)** *Suppose* $\|X - L_0 - S_0\|_F \le \varepsilon$. *Let* $A \in \mathbb{R}^{m \times d}$ *with SVD* $A = U_A \Sigma_A V_A^T$ *be a column-wisely unit-normed dictionary matrix which satisfies* $\mathcal{P}_{U_A}(U_0) = U_0$ *(i.e.,* $U_0$ *is a subspace of* $U_A$*). For any* $0 < \epsilon < 0.35$ *and some numerical constant* $c_a > 1$, *if*

$$\mathrm{rank}\,(L_0) \le \mathrm{rank}\,(A) \le \frac{\epsilon^2 n_2}{c_a \mu_1(A) \log n_1} \quad and \quad |\Omega| \le (0.35 - \epsilon)mn, \qquad (3.10)$$

*then with probability at least* $1 - n_1^{-10}$, *any solution* $(Z^*, S^*)$ *to (3.9) with* $\lambda = 1/\sqrt{n_1}$ *gives a near recovery to* $(L_0, S_0)$, *in a sense that* $\|AZ^* - L_0\|_F \le 8\sqrt{mn}\varepsilon$ *and* $\|S^* - S_0\|_F \le (8\sqrt{mn} + 2)\varepsilon$.

### 3.3 An Unsupervised Algorithm for Matrix Recovery

To handle coherent (equivalently, less incoherent) data, Theorem 1 suggests that the dictionary matrix $A$ should be low-rank and satisfy $U_0 \subset U_A$. In certain supervised environment, this might not be difficult as one could potentially use clear, well processed training data to construct the dictionary. In an unsupervised environment, however, it will be challenging to identify a low-rank dictionary that obeys $U_0 \subset U_A$. Note that $U_0 \subset U_A$ can be viewed as supervision information (if $A$ is low-rank).

In this paper, we will introduce a heuristic algorithm that can work distinctly better than RPCA in an unsupervised environment. As can be seen from (3.7), RPCA is actually not brittle with respect to coherent data (although its performance is depressed). Based on this observation, we propose a simple algorithm, as summarized in Algorithm 1, to achieve a solid improvement over RPCA. Our idea is straightforward: We first obtain an estimate of $L_0$ by using RPCA and then utilize the estimate to construct the dictionary matrix $A$ in LRR. The post-processing steps (Step 2 and Step 3) that slightly modify the solution of RPCA is to encourage well-conditioned dictionary, which is the circumstance favoring LRR.

Whenever the recovery produced by RPCA is already exact, the claim in Theorem 1 gives that the recovery produced by our Algorithm 1 is exact as well. That is to say, in terms of exactly recovering $L_0$ from a given $X$, the success probability of our Algorithm 1 is greater than or equal to that of RPCA. From the computational perspective, Algorithm 1 does not really double the work of RPCA, although there are two convex programs in our algorithm. In fact, according to our simulations, usually the computational time of Algorithm 1 is merely about 1.2 times as much as RPCA. The reason is that, as has been explored by [13], the complexity of solving the LRR problem (1.3) is $O(n^2 r_A)$ (assuming $m = n$), which is much lower than that of RPCA (which requires $O(n^3)$) provided that the obtained dictionary matrix $A$ is fairly low-rank (i.e., $r_A$ is small).

One may have noticed that the procedure of Algorithm 1 could be made iterative, i.e., one can consider $\hat{A}Z^*$ as a new estimate of $L_0$ and use it to further update the dictionary matrix $A$, and so on. Nevertheless, we empirically find that such an iterative procedure often converges within two iterations. Hence, for the sake of simplicity, we do not consider iterative strategies in this paper.

---

**Algorithm 1** Matrix Recovery

---

**input:** Observed data matrix $X \in \mathbb{R}^{m \times n}$.

**adjustable parameter:** $\lambda$.

**1.** Solve for $\hat{L}_0$ by optimizing the RPCA problem (1.2) with $\lambda = 1/\sqrt{n_1}$.

**2.** Estimate the rank of $\hat{L}_0$ by

$$\hat{r}_0 = \#\{i : \sigma_i > 10^{-3}\sigma_1\},$$

where $\sigma_1, \sigma_2, \cdots, \sigma_{n_2}$ are the singular values of $\hat{L}_0$.

**3.** Form $\tilde{L}_0$ by using the rank-$\hat{r}_0$ approximation of $\hat{L}_0$. That is,

$$\tilde{L}_0 = \arg\min_L \|L - \hat{L}_0\|_F^2, \text{ s.t. rank}(L) \le \hat{r}_0,$$

which is solved by SVD.

**4.** Construct a dictionary $\hat{A}$ from $\tilde{L}_0$ by normalizing the column vectors of $\tilde{L}_0$:

$$[\hat{A}]_{:,i} = \frac{[\tilde{L}_0]_{:,i}}{\|[\tilde{L}_0]_{:,i}\|_2}, i = 1, \cdots, n,$$

where $[\cdot]_{:,i}$ denotes the $i$th column of a matrix.

**5.** Solve for $Z^*$ by optimizing the LRR problem (1.3) with $A = \hat{A}$ and $\lambda = 1/\sqrt{n_1}$.

**output:** $\hat{A}Z^*$.

---

## 4 Experiments

### 4.1 Results on Randomly Generated Matrices

We first verify the effectiveness of our Algorithm 1 on randomly generated matrices. We generate a collection of $200 \times 1000$ data matrices according to the model of $X = \mathcal{P}_{\Omega^\perp}(L_0) + \mathcal{P}_{\Omega}(S_0)$: $\Omega$ is a support set chosen at random; $L_0$ is created by sampling 200 data points from each of 5 randomly generated subspaces; $S_0$ consists of random values from Bernoulli $\pm 1$. The dimension of each subspace varies from 1 to 20 with step size 1, and thus the rank of $L_0$ varies from 5 to 100 with step size 5. The fraction $|\Omega|/(mn)$ varies from 2.5% to 50% with step size 2.5%. For each pair of rank and support size $(r_0, |\Omega|)$, we run 10 trials, resulting in a total of 4000 ($20 \times 20 \times 10$) trials.

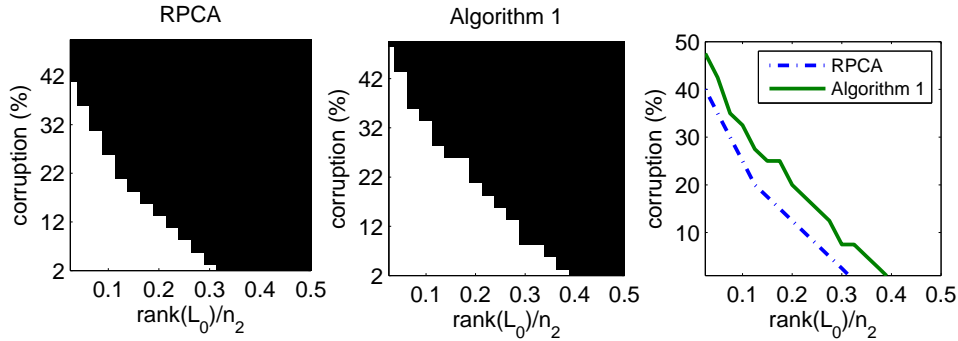

Figure 4: Algorithm 1 vs RPCA for the task of recovering randomly generated matrices, both using $\lambda = 1/\sqrt{n_1}$. A curve shown in the third subfigure is the boundary for a method to be successful — The recovery is successful for any pair $(r_0/n_2, |\Omega|/(mn))$ that locates below the curve. Here, a success means $\|\hat{L}_0 - L_0\|_F < 0.05\|L_0\|_F$, where $\hat{L}_0$ denotes an estimate of $L_0$.

Figure 4 compares our Algorithm 1 to RPCA, both using $\lambda = 1/\sqrt{n_1}$. It can be seen that, using the learned dictionary matrix, Algorithm 1 works distinctly better than RPCA. In fact, the success area (i.e., the area of the white region) of our algorithm is 47% wider than that of RPCA! We should also mention that it is possible for RPCA to be exactly successful on coherent (or less incoherent) data, provided that the rank of $L_0$ is low enough and/or $S_0$ is sparse enough. Our algorithm in general improves RPCA when $L_0$ is moderately low-rank and/or $S_0$ is moderately sparse.

## 4.2 Results on Corrupted Motion Sequences

We now present our experiment with 11 additional sequences attached to the Hopkins155 [21] database. In those sequences, about 10% of the entries in the data matrix of trajectories are unobserved (i.e., missed) due to vision occlusion. We replace each missed entry with a number from Bernoulli $\pm 1$, resulting in a collection of corrupted trajectory matrices for evaluating the effectiveness of matrix recovery algorithms. We perform subspace clustering on both the corrupted trajectory matrices and the recovered versions, and use the clustering error rates produced by existing subspace clustering methods as the evaluation metrics. We consider three state-of-the-art subspace clustering methods: Shape Interaction Matrix (SIM) [5], Low-Rank Representation with $A = X$ [14] (which is referred to as "LRRx") and Sparse Subspace Clustering (SSC) [6].

Table 1: Clustering error rates (%) on 11 corrupted motion sequences.

|  | Mean | Median | Maximum | Minimum | Std. | Time (sec.) |
|---|---|---|---|---|---|---|
| SIM | 29.19 | 27.77 | 45.82 | 12.45 | 11.74 | 0.07 |
| RPCA + SIM | 14.82 | 8.38 | 45.78 | 0.97 | 16.23 | 9.96 |
| Algorithm 1 + SIM | 8.74 | 3.09 | 42.61 | 0.23 | 12.95 | 11.64 |
| LRRx | 21.38 | 22.00 | 56.96 | 0.58 | 17.10 | 1.80 |
| RPCA + LRRx | 10.70 | 3.05 | 46.25 | 0.20 | 15.63 | 10.75 |
| Algorithm 1 + LRRx | 7.09 | 3.06 | 32.33 | 0.22 | 10.59 | 12.11 |
| SSC | 22.81 | 20.78 | 58.24 | 1.55 | 18.46 | 3.18 |
| RPCA + SSC | 9.50 | 2.13 | 50.32 | 0.61 | 16.17 | 12.51 |
| Algorithm 1 + SSC | 5.74 | 1.85 | 27.84 | 0.20 | 8.52 | 13.11 |

Table 1 shows the error rates of various algorithms. Without the preprocessing of matrix recovery, all the subspace clustering methods fail to accurately categorize the trajectories of motion objects, producing error rates higher than 20%. This illustrates that it is important for motion segmentation to correct the gross corruptions possibly existing in the data matrix of trajectories. By using RPCA ($\lambda = 1/\sqrt{n_1}$) to correct the corruptions, the clustering performances of all considered methods are improved dramatically. For example, the error rate of SSC is reduced from 22.81% to 9.50%. By choosing an appropriate dictionary for LRR ($\lambda = 1/\sqrt{n_1}$), the error rates can be reduced again, from 9.50% to 5.74%, which is a 40% relative improvement. These results verify the effectiveness of our dictionary learning strategy in realistic environments.

## 5 Conclusion and Future Work

We have studied the problem of disentangling the low-rank and sparse components in a given data matrix. Whenever the low-rank component exhibits clustering structures, the state-of-the-art RPCA method could be less successful. This is because RPCA prefers incoherent data, which however may be inconsistent with data in the real world. When the number of clusters becomes large, the second and third coherence parameters enlarge and hence the performance of RPCA could be depressed. We have showed that the challenges arising from coherent (equivalently, less incoherent) data could be effectively alleviated by learning a suitable dictionary under the LRR framework. Namely, when the dictionary matrix is low-rank and contains information about the ground truth matrix, LRR can be immune to the coherence parameters that increase with the underlying cluster number. Furthermore, we have established a practical algorithm that outperforms RPCA in our extensive experiments.

The problem of recovering coherent data essentially concerns the robustness issues of the Generalized PCA (GPCA) [22] problem. Although the classic GPCA problem has been explored for several decades, robust GPCA is new and has not been well studied. The approach proposed in this paper is in a sense preliminary, and it is possible to develop other effective methods for learning the dictionary matrix in LRR and for handling coherent data. We leave these as future work.

## Acknowledgement

Guangcan Liu was a Postdoctoral Researcher supported by NSF-DMS0808864, NSF-SES1131848, NSF-EAGER1249316, AFOSR-FA9550-13-1-0137, and ONR-N00014-13-1-0764. Ping Li is also partially supported by NSF-III1360971 and NSF-BIGDATA1419210.

## Footnotes

[1]It is not crucial to determine the exact value of $d$. Suppose $Z^*$ is the optimal solution with respect to $Z$. Then LRR uses $AZ^*$ to restore $L_0$. LRR falls back to RPCA when $A = \mathtt{I}$ (identity matrix). Furthermore, it can be proved that the recovery produced by LRR is the same as RPCA whenever the dictionary $A$ is orthogonal.

[2]In this paper, SVD always refers to skinny SVD. For a rank-$r$ matrix $M \in \mathbb{R}^{m \times n}$, its SVD is of the form $U_M \Sigma_M V_M^T$, with $U_M \in \mathbb{R}^{m \times r}$, $\Sigma_M \in \mathbb{R}^{r \times r}$ and $V_M \in \mathbb{R}^{n \times r}$.

[3]In terms of *exact* recovery, $O(n_2 / \log n_1)$ is probably the "finest" bound one could accomplish in theory.

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
