[Reviews · NeurIPS 2014]

Submitted by Assigned_Reviewer_2

This paper addresses the problem of robustly estimating the low-dimensional subspace of contaminated observations when the observations are inherently coherent. Performance goes worse with increasing data coherence is a standard theoretical bottleneck of previous RPCA methods. This paper, however, circumvents this problem in a clever manner. Considering that such cluster structure is rather common in realistic data, solving this issue is certainly significantly meaningful. The proposed method is both theoretically sound and well demonstrated to perform well in practice.

I just have a curious comment on the submission:

Can the proposed method also be applied for solving subspace clustering/segmentation problem? Could the authors provide some comments on this point?

Summary: This paper solves an important problem in robust PCA works. The quality of the paper is very good and should be accepted.

Submitted by Assigned_Reviewer_26

This paper combines two recent techniques, robust PCA and
dictionary-based structure-preserving projection, in the task of
restoring corrupted observartion matrices. The key insight is that
for structured (for instance clustered) data, the guarantees of robust
PCA are not strong enough, and by representing the structure with
by a dictionary, stronger bounds can be given. Theorems are given to
support these claims. The remaining problem is how to learn the
dictionary. For that an algorithm justified by empirical results is
given: First compute robust PCA, then represent the result by a
dictionary.

Quality:

The paper is technically sound, up to a reasonable level of
checking. The two theorems elaborate conditions under which
reconstruction is possible, given a dictionary. For the learning of
the dictionary, the justification is empirical.

Clarity:

Clarity is the main problem of the paper. While the structure is clear,
the main contributions and impact of the paper have not been explained
clearly enough to be accessible beyond a very narrow specialist audience.
The claims would need to be more well-defined, and the impact of the
results explicated. For instance, the concept coherence is hard to understand,
and means of interpreting it are very indirect (and given in the appendix).
One of the potential impacts of the paper may be that by using
dictionaries many earlier problems can be sidestepped.

Also the language needs checking.

It would be important to get a comment from the authors about what can be
done about these issues.

Originality:

The paper is based on a combination of recent techniques, but includes
new theorems and empirical results supporting the usefulness of the
combination.

Significance:

The paper does advance the state-of-the art with rigorous
results. Even though the proposed algorithm for finding the dictionary
is heuristic, clear improvements over alternative algorithmns
demonstrate that the insights given this paper are useful.

---
Comment after the author rebuttal and reviewer discussion:

There is clearly publishable content and interesting contributions. The only remaining concern I have is that it will be hard work to re-write the paper to make it accessible to very specialists. If there was an option for a "major revision" I would vote for it.

Summary: The paper combines two recent ideas, and gives both theoretical
results and a heuristic but empirically well-performing algorithm.

Submitted by Assigned_Reviewer_39

The paper proposes to use Low Rank Representation (LRR) with some learned dictionary A to improve the effectiveness of RPCA. In particular, the authors show that the incoherence parameter \mu which often thought of as a bottleneck in the recovery guarantee in RPCA can in fact correspond to some additional clustering structure in the data. Under some condition on the dictionary A, the authors show that one can exploit such structure using LRR and partially remove the dependence on \mu and get a theoretically stronger guarantee in both exact low-rank and sparse decomposition and its noisy extension.
The method is novel and significant to the field. It is closely related to the union-of-subspace structure assumed in previous subspace clustering papers such as “Wang and Xu: Noisy Sparse Subspace Clustering & Provable subspace clustering: When LRR meets SSC”, and “Soltanolkotabi and Candes: A geometric analysis of subspace clustering with outliers” “Soltanolkotabi et al: Robust subspace clustering” but are less explicit about the assumption on “clustering”, which is reasonable and more probably more general. The simulation and real data experiments verify the theoretical analysis and I think the new structure can be found in many other real applications too (such as text data as remarked in the paper).
In summary, the paper contains a substantial contribution to the field of compressed sensing and I think it should be accepted by NIPS.
That said, I do have a number of discussions and stylistic suggestions which are summarized in the detailed comments below.

1. Line 80-82: Need some re-writing to make the description clearer. What do you mean by “our data”? Also even if the problem captures all the structures in the problem, there is still yet another condition to qualify to have “perfect recovery”.
2. Line 88: … is no longer a method of perfect (WHAT?).
3. Please proof-read the papers more carefully for problems like the above two. Be specific and precise about what to claim and what you are referring to in the text. In general, please work on clarity of the language in the paper.
4. Wang and Xu also have written in their paper “Noisy Sparse Subspace Clustering” how higher coherence parameter is actually good for subspace clustering (unlike in RPCA and matrix completion). The key difference from here is that the data need not be low-rank in overall, only every cluster has to be low-rank. The structure exploited here in this paper can be thought of as a combination of low-rank structure and union-of-subspace cluster structure.
5. It will be interesting to compare the proposed algorithm against first solving noisy subspace clustering with \ell_1 penalty then do PCA for each subspace. No provable guarantee exists to date for the later version though.
6. Section 4.2 essentially proposes a new method for solving subspace clustering problem with sparse corruption in the data. The method leads to significant improvements over the standard RPCA+SSC on Hopkins155 dataset which I think is a hidden contribution here in this paper and the authors should point it out.
Summary: This paper should be accepted because it contains at least three important contributions to the field including: 1. descriptions of a clustering structure that leads to high coherence, 2. example that one can implicitly exploit such structure by using a dictionary in LRR, 3. partially solve the sparse corruptions problem in subspace clustering problem.
Author Feedback
Author rebuttal: To Reviewer_2:

Many thanks for your very positive comments. Regarding the question that "Can the proposed method also be applied for solving subspace clustering/segmentation problem?", the answer is "Yes". Section 4.2 has indeed validated the effectiveness of the proposed method as a new procedure for subspace clustering. We shall more explicitly point out this contribution in the final version. Reviewer_39 kindly commented this as a "hidden contribution".

----------------------
To Reviewer_26:

Many thanks for supporting this paper. We appreciate your concern that the concept of coherence is not easy to follow. We included a two-page section in the appendix (supplementary material) to elaborate on the concept of coherence. We will make even more effort to try to make the concept of coherence to a broad (NIPS) audience, in the main body of the paper.

The difficulty in understanding coherence is partly due to the fact that the mathematical definition of coherence is very abstract (because the definition is based on SVD) and, what is more, there is a lack of studies on the physical meaning of coherence parameters. For the first time, this paper investigates the physical meaning of coherence. Namely, we show that the coherence parameters are actually characterizing some clustering structures widely existed in data. So, as pointed out in Introduction, one contribution of our work is indeed to help understand the concept of coherence. Our work also reveals the actuality that coherence is indeed an important concept and should become well-known to broad audiences in near future, although only the theoretical community knows well this concept in the current stage.

Again, we will make more effort in the main body of the paper to better clarify the definition of coherence. The appendix (which has space for thorough elaborations) will be posted online as well. Also, we will add more discussions in Section 3.1 and Appendix to better clarify the concept of coherence.

We probably should iterate that the contributions of this paper are actually (far) more than "a combination of recent techniques". As pointed out by Reviewer_39,

"... the authors show that the incoherence parameter $\mu$ which often thought of as a bottleneck in the recovery guarantee in RPCA can in fact correspond to some additional clustering structure in the data. Under some condition on the dictionary $A$, the authors show that one can exploit such structure using LRR and partially remove the dependence on $\mu$ and get a theoretically stronger guarantee in both exact low-rank and sparse decomposition and its noisy extension..."

Therefore, one original contribution of this paper is the proposal of a methodology that can break through the bottleneck of the incoherent condition commonly assumed by previous work. Thank you very much.

---------------------
To Reviewer_39:

Many thanks for your very positive comments and the pointers to several relevant papers. They will be included in the references.

Q1: Line 80-82: Need some re-writing to make the description clearer. What do you mean by “our data”? Also even if the problem captures all the structures in the problem, there is still yet another condition to qualify to have “perfect recovery”.
Line 88: … is no longer a method of perfect (WHAT?). Please proof-read the papers more carefully for problems like the above two. Be specific and precise about what to claim and what you are referring to in the text. In general, please work on clarity of the language in the paper.

R1: Thanks a lot. We will carefully refine the language to make sure the clarity of writing.

Q2: Wang and Xu also have written in their paper “Noisy Sparse Subspace Clustering” how higher coherence parameter is actually good for subspace clustering (unlike in RPCA and matrix completion). The key difference from here is that the data need not be low-rank in overall, only every cluster has to be low-rank. The structure exploited here in this paper can be thought of as a combination of low-rank structure and union-of-subspace cluster structure.

R2: This is a very interesting point. We will include Wang and Xu's work for discussion in the final version of this paper. We have indeed tested SSC (noisy version) in Section 4.2. Without the preprocessing of matrix recovery methods, as shown in Table 1, SSC (with the best parameter) produces a high error rate of 22.81\%. Yet, this result actually does not conflict with Wang and Xu's work, which states that (noisy) SSC needs to assume the magnitude of noise and thus may not solve the cases where the corruptions have large magnitudes. While producing the final version, we will make this point be clear in Section 4.2.

Q3: It will be interesting to compare the proposed algorithm against first solving noisy subspace clustering with \ell_1 penalty then do PCA for each subspace. No provable guarantee exists to date for the later version though.

R3: Thanks for pointing out this. We shall explore the potential subspace clustering method (which is actually new) you mentioned in future work.

Q4: Section 4.2 essentially proposes a new method for solving subspace clustering problem with sparse corruption in the data. The method leads to significant improvements over the standard RPCA+SSC on Hopkins155 dataset which I think is a hidden contribution here in this paper and the authors should point it out.

R4: Yes, we shall explicitly point out this contribution in the final version.